# *CLN8* Gene Compound Heterozygous Variants: A New Case and Protein Bioinformatics Analyses

**DOI:** 10.3390/genes13081393

**Published:** 2022-08-05

**Authors:** Rajech Sharkia, Abdelnaser Zalan, Hazar Zahalka, Amit Kessel, Ayman Asaly, Wasif Al-Shareef, Muhammad Mahajnah

**Affiliations:** 1Unit of Human Biology and Genetics, The Triangle Regional Research and Development Center, Kafr Qara 30075, Israel; 2Beit-Berl Academic College, Beit-Berl 4490500, Israel; 3Child Neurology and Development Center, Hillel-Yaffe Medical Center, Hadera 38100, Israel; 4Department of Biochemistry and Molecular Biology, Faculty of Life Sciences, Tel-Aviv University, Tel-Aviv 69978, Israel; 5Pediatrics Department, Hillel-Yaffe Medical Center, Hadera 38100, Israel; 6Rappaport Faculty of Medicine, Technion-Israel Institute of Technology, Haifa 3200003, Israel

**Keywords:** neuronal ceroid lipofuscinoses (NCL), CLN8 disease, *CLN8* gene, compound heterozygous variant, neurodegenerative disease, bioinformatic analyses

## Abstract

The CLN8 disease type refers to one of the neuronal ceroid lipofuscinoses (NCLs) which are the most common group of neurodegenerative diseases in childhood. The clinical phenotypes of this disease are progressive neurological deterioration that could lead to seizures, dementia, ataxia, visual failure, and various forms of abnormal movement. In the current study, we describe two patients who presented with atypical phenotypic manifestation and protracted clinical course of CLN8 carrying a novel compound heterozygous variant at the *CLN8* gene. Our patients developed a mild phenotype of CLN8 disease: as they presented mild epilepsy, cognitive decline, mild learning disability, attention-deficit/hyperactivity disorder (ADHD), they developed a markedly protracted course of motor decline. Bioinformatic analyses of the compound heterozygous *CLN8* gene variants were carried out. Most of the variants seem likely to act by compromising the structural integrity of regions within the protein. This in turn is expected to reduce the overall stability of the protein and render the protein less active to various degrees. The cases in our study confirmed and expanded the effect of compound heterozygous variants in CLN8 disease.

## 1. Introduction

Neuronal ceroid lipofuscinosis (NCL) refers to a group of rare disorders that affects human neurons which is one of the most common causes of progressive neurological deterioration in childhood. It involves the buildup of an abnormal material called lipofuscin in the brain. NCL is inherited from the parents by their progeny in an autosomal recessive manner. There are three main types of NCL namely, Adult (Kufs or Parry disease), Juvenile (Batten disease), and Late infantile (Jansky–Bielschowsky disease) [1]. The late infantile type is the most heterogeneous one, with several variants and four genes identified: *CLN2*, *CLN5*, *CLN6*, and *CLN8* [2].

The accumulation of autofluorescent lysosomal storage material in the central nervous system is a key pathological finding of NCL. Several possible candidate genes (13 genes) are involved in this process, namely, *CLN1* to *CLN8*, and *CLN10* to *CLN14* [3]. To date, more than 430 pathogenic variants in the above 13 candidate genes have been reported in human NCLs, and most have been registered in the NCL Mutation Database [4,5,6]. As a collective group, NCL patients are typically defined by observation of cognitive and visual impairments, epileptic seizures, and deterioration of motor skills and balance issues [7]. Unfortunately, there is no cure for NCL disorders. Therapeutic approaches for the treatment of many NCLs include the administration of immunosuppressive agents to antagonize neuroinflammation associated with neurodegeneration, the use of various small molecules, stem cell therapy, and gene therapy [8].

The CLN8 disease type is clinically recognized during childhood in which two main clinical phenotypes were described, namely “Northern Epilepsy” known as the Finnish type characterized by intractable epilepsy and cognitive regression at the ages 5–10 years [9]. The second phenotype was the late–infantile NCL characterized by earlier onset and more rapid progressive course with visual failure, ataxia, mental decline, and epilepsy [10,11].

At least 25 mutations in the *CLN8* gene have been found to cause CLN8 disease [12]. These mutations in the *CLN8* gene are implicated in nervous system pathologies [13,14]. The product of the *CLN8* gene is a non-glycosylated ER membrane protein. It contains the TLC (TRAM-LAG1-CLN8) domain, involved in the homeostasis of brain lipids: synthesis, sensing, transport, and metabolism [15]. However, it is unclear how exactly the protein carries out its molecular function, on both the sequence and structure levels. The three-dimensional structure of CLN8 has not been determined so far.

Locally, the first case of CLN8 disease in an Israeli child was described in 2007 [10]. The patient was a 5-year-old male who presented with epilepsy and gait ataxia. Exon sequencing revealed a homozygous missense mutation (766C > G, p.(Gln256Glu)) at the *CLN8* gene (NM_018941.3). Since then, three more patients were identified, all homozygous for the same mutation but with a phenotypic heterogeneity [16].

In the current study, we report two female siblings presented with atypical phenotypic manifestation and protracted clinical course of CLN8 carrying a novel compound heterozygous variant, (c.473A > G, p.(Tyr158Cys)) and (766C > G, p.(Gln256Glu)), at the *CLN8* gene. Additionally, bioinformatics analyses of the *CLN8* gene variants were carried out and their consequences on the protein structure were discussed.

## 2. Materials and Methods

### 2.1. Clinical Description

This research was approved by the ethics committee of the Hillel-Yaffe Medical Center, in the city of Hadera in Israel. Two sisters suffering from epilepsy who belong to a family in a village from the Arab society of Israel were referred to the pediatric neurology clinic. Their parents were healthy and distant relatives. The exact relationship between the parents is that the mother’s grandfather and the father’s grandmother are a brother and sister (see pedigree—Figure 1). It is noteworthy that in our previous study, other relatives had been diagnosed with a mutation in the *CLN8* gene [16]. Moreover, another unrelated family living in the same village had two siblings diagnosed with a *CLN8* gene variant. According to these data, we hypothesized that a variant in *CLN8* gene could be the cause for this disease in these two patients.

#### 2.1.1. Patient A

The elder one of the two siblings was a 14-year-old girl who was born after a normal pregnancy. Her early development was unremarkable except for attention-deficit hyperactivity disorder (ADHD) that was diagnosed at the age of 6 years with efficient methylphenidate treatment. At the age of 8 years and 5 months she was admitted to the pediatrics department due to two generalized tonic-clonic seizures. Initial EEG test and brain MRI were normal, anti-epileptic therapy with valproic acid was initiated. At follow up she started having absence seizures, video EEG was normal, while a repeated brain imaging demonstrated mild cerebellar atrophy. After a period of four months, she started to have learning difficulties and behavioral problems at school with declining academic achievements in the following years. As she reached 11 years of age, sleep onset seizures appeared, neurological examination revealed new signs of limitation in ophthalmic smooth pursuit bilaterally without visual impairment. MRI showed a 7 mm signal change abnormality in the left thalamus and cerebellar atrophy. Several months later her tandem gait became unsteady, her gait worsened with time. At the age of 13 years, full ataxic gait was observed during her neurological examination. Her detailed clinical manifestations are described in Table 1.

#### 2.1.2. Patient B

The younger affected sister, 12 years old, was born after a normal pregnancy and delivery. She achieved early developmental milestones at appropriate ages. When she reached the age of 7 years, she was diagnosed with learning disability and ADHD. Later, her first seizure appeared at the age of 7.5 years, the seizure was described as tonic-clonic convulsion. Her EEG was without abnormal findings, ophthalmological examination was also normal, while the brain MRI revealed mild cerebellar atrophy. Subsequently, she had frequent seizures and was diagnosed with epilepsy, therefore, treatment with Valproic acid was initiated; thus, she had convulsion regression. Several months later, through neurological follow-up, it was noticed that she had difficulty with tandem walking; however, up to the date of preparing this report she had not developed ataxia or other neurological deficits. The detailed clinical manifestations of this patient are also described in Table 1.

### 2.2. Genetic Analysis

Genomic DNA was extracted from blood samples of all family members using standard procedures. To identify the causative variants of *CLN*, Whole Exome Sequencing (WES) was performed initially for the two affected daughters. To confirm the identified WES variants, genomic regions of interests (NM_018941.3) were verified by Sanger sequencing. The target *CLN8* exons, in the two affected daughters analyzed by WES, and in the other family members (the two parents and the two normal daughters) were amplified.

### 2.3. Structural Bioinformatic Analysis

Structure prediction was carried out using AlphaFold, a machine learning algorithm that was developed in recent years, and which predicts structures at near-experimental accuracy [17,18]. Using deep networks, AlphaFold finds evolutionary correlations within amino acid positions in the query protein and its homologues, and these correlations are then used for modeling the three-dimensional structure of the query protein. Moreover, the processed data is used also for estimating the accuracy of the prediction. The change in stability of the modeled protein due to mutations was predicted by the DUET webserver [19]. By using Support Vector Machines (SVMs), this method integrates different computational approaches to reach a consensus prediction. The per-residue evolutionary conservation of the CLN8 protein was calculated and mapped onto the protein’s modeled structure using the ConSurf webserver [20], based on the phylogenetic relations between the protein and its homologous sequences. The calculation was run with default values, 232 homologues, and the empirical Bayesian method. The highly popular Dali webserver was used to heuristically search for structural analogues of the query protein in the entire Protein Data Bank [21,22].

## 3. Results

To reveal the genetic basis of the diseases, the *CLN8* targeted gene was tested for the two patients. It was confirmed that they are carriers to the missense mutation (c.766C > G, p.(Gln256Glu)) that was previously identified in that village which they inherited maternally. Sequencing of *CLN8* gene revealed the presence of an additional variant (c.473A > G, p.(Tyr158Cys)), which was interpreted as pathogenic according to (ClinVar—US library of medicine) and inherited paternally in our patients. The mother and father were carriers for a single variant each and had no symptoms or signs of neurological deficits. A detailed description of the relationship between the current cases and the cases of our previous studies was presented in the pedigree (Figure 1). We concluded that these compound variants in the *CLN8* gene are most likely the cause of this atypical clinical presentation of CLN8 disease.

### 3.1. Mutational Analysis of the CLN8 Gene Product

To gain insights about the possible mechanisms of the variants, we predicted the structure of the protein using the state-of-the-art algorithm AlphaFold [17,18], and obtained a model with very high prediction confidence (Figure 2). This model, as well as other bioinformatics tools, were then used to analyze the variants, as follows:

#### 3.1.1. Variants Found by Our Study

I. Q256E (c.766C > G)

The variant renders Q256 negatively charged, while keeping its size almost unchanged. Since the residue faces the inside of the protein, its charging is likely to destabilize the protein due to desolvation of the charged glutamate. This is supported by calculations of stability changes, carried out by the DUET webserver [19]. The destabilization is likely to induce conformational changes in the vicinity of position 256, which may spread throughout the protein and lead to inhibition of its activity. The variant may also act more directly (i.e., not through conformational changes), by disrupting a specific biological function of Q256. Calculations of the evolutionary conservation of CLN8, carried out by the ConSurf webserver [20], show Q256 and its immediate surrounding region of the protein to be highly conserved (Figure 3). This suggests that the function of the protein might involve this region, and thus, a variant-induced replacement of Q256 may disrupt this function. A scan of CLN8 against the Protein Data Bank (PDB) [21] did not reveal any homologous proteins of known structure, which could contribute functional information about CLN8. However, by using the Dali webserver [22], we found the following two proteins that, despite being unrelated to CLN8, were structurally similar to the CLN8 protein (r.m.s.d of ~4.6 Å), and whose structures were determined with bound ligands:The TACAN mechanosensitive ion channel, bound to coenzyme A (PDB ID: 7n0l, sequence identity to CLN8: 14%).ELOVL fatty acid elongase 7, bound to a coenzyme A-fatty acid conjugate (PDB ID: 6y7f, sequence identity to CLN8: 12%).

Interestingly, when the two structures are superposed on CLN8, Q256 faces one of the coenzyme A phosphate groups in the ligands (Figure 4A,B). Since CLN8 is functionally associated with lipids, including phospholipids, it is possible that Q256 interacts favorably with a phosphate group in the protein’s natural ligand or substrate. If so, the Q256E variant is expected to render this interaction unfavorable due to the electrostatic repulsion between the negatively charged glutamate and phosphate groups.

II. Y158C (c.473A > G)

Y158 is evolutionarily non-conserved but it forms several non-covalent interactions with nearby residues (Figure 5). This includes a hydrogen bond with H210 and π-cation interactions with K95. These interactions are likely to be lost with the replacement of Y158 by cysteine, which is shorter and incapable of forming π interactions. However, since the residues involved in the interactions are also non-conserved, it is difficult to infer their importance and, therefore, if the loss of their interactions with position 158 is likely to have a functional effect. 

#### 3.1.2. Mutations reported by Cannelli et al.’s Study in 2006

I. Deletion After Position 22 (c.66delG)

This mutation deletes a large part of the protein and is therefore expected to result in a nonfunctional protein molecule, which most likely does not even reach the ER membrane.

II. Y158C (c.473A > G)

See Section 3.1.1 II above.

III. Q194R (c.581A > G)

Q194 is an evolutionarily conserved residue. It is located in the region of the protein predicted above to be functionally important, and which includes also Q256. Q194, along with other highly conserved residues, forms a network of non-covalent interactions that stabilize this region and keep helices 157–182, 188–216, and 229–258, close together (Figure 6). The residues involved in this network have small- or medium-size side chains and are tightly packed together. Thus, replacement of Q194 with a large arginine residue is expected to create significant atomic clashes within the helical bundle, compromising its structural integrity. This effect would be further exacerbated by the desolvation of the electrically charged side chain of arginine.

#### 3.1.3. Mutation Reported by Allen et al.’s Study in 2012, Deletion after Position 188 (c.562_563delCT)

This mutation deletes a large part of the protein and is therefore expected to result in a nonfunctional protein molecule, which most likely does not even reach the ER membrane.

#### 3.1.4. Mutations Reported by Beesley et al.’s Study in 2016

I. Polypeptide Chain Termination After Position 255 (c.763C > T)

The presence of a nonsense mutation at position c.763C > T (codon 255) leads to premature chain termination. Accordingly, all the following amino acids (codon 256–286) will not be synthesized. Most importantly, the codons 283–286 (Lys-Lys-Arg-Pro; ER-retrieval signal) will not be expressed. Therefore, the mutant will not reach the ER.

II. L243P (c.728T > C)

L243 is located in the middle of the same α-helix that contains also Q256 (see Figure 2 and Figure 3). Replacement of L243 with proline is expected to create a kink in the helix. The resulting conformational change might disrupt the activity of the protein by weakening the favorable interactions between the helix and adjacent helices, as well as with the protein’s ligand (e.g., via Q256, see above).

#### 3.1.5. Mutation Reported by Sanchez et al.’s Study in 2016, A67V (c.200C > T)

A67 is located on helix 58–87. Its side forms tight nonpolar interactions with several residues that reside on helix 157–182, such as W177 and C174 (Figure 7). These interactions stabilize the locations of the two helices with respect to one another. The A67V mutation is expected to result in displacements of the two helices due to the introduction of a larger, branched side chain at position 67. Helix 157–182 faces a cavity in the protein, which we suggested above to serve as a possible ligand/substrate-binding site, based on structural analogues of the protein (see analysis of the Q256E mutation). Thus, its mutation-induced displacement might lead to the loss of favorable interactions between the putative ligand/substrate and helix 157–182 residues, such as S169, T173, and S176, leading to reduced biological activity of the protein. 

#### 3.1.6. Mutation Reported by Gao et al.’s Study in 2018

I. Q100T (c.298C > T)

Q100 is located far from the putatively functional region of CLN8, which contains Q256 (see Figure 3). Q100 itself is only moderately conserved. However, it forms a hydrogen bond with D88, which is evolutionarily conserved (Figure 8). The hydrogen bond is one of the non-covalent interactions that keep helices 58–87 and 101–128 close to each other. Since these helices span the entire length of the protein, interact with other helices, and contain evolutionarily conserved resides, it is likely that disrupting the interaction between them will affect large parts of the protein, and diminish the activity of the protein. The Q100T mutation replaces one polar residue with another, both are capable of forming hydrogen bonds. However, threonine is shorter than glutamine and may be unable to form the hydrogen bond with D88.

## 4. Discussion

It was previously known that the disease of CLN8 had two main phenotypes. The first one namely the Northern Epilepsy which is characterized by a distinctive progressive myoclonic epilepsy described in Finland [9]. The second variant is late-infantile NCL which was described in Turkey [23]. In one of our previous studies, it was found that the variant late–infantile NCL was characterized by earlier onset and a more rapid progressive course with visual failure, ataxia, mental decline, and epilepsy. A homozygous missense mutation in the *CLN8* gene (766C > G, p.(Gln256Glu)) was found to be responsible for these cases in one village [10]. Consequently, three more patients, who are distantly related to the previous patients and carry the same homozygous mutations but with a phenotypic heterogeneity, were identified in the same village [16].

Here, we report a new case study of two patients (female siblings) who have atypical clinical features of CLN8 disease. Noticeably, these two patients were relatives to our previous study cases from the same Arab village in Israel. Compared to the other cases, which had a homozygous variant in *CLN8* gene, the current cases were diagnosed to have a novel compound heterozygous variant at the same gene.

Most of the variants in the *CLN8* gene that cause the disease were found to be homozygous variants, while few reports were recorded to have heterozygous ones which are presented in Table 1. These cases were noted to have varying clinical manifestations that range from symptoms appearing since birth to milder forms at the age of 4 years.

The mean age of the disease onset, according to the five studies [24,25,26,27,28] presented in Table 1, is about 4 years, while our two patients had late onset age of 7.5 and ~8.5 years. This late onset of the disease in our cases was reflected in various clinical manifestations such as abnormal gait, visual impairment, and epilepsy. On the other hand, it was noted that both of our patients did not have myoclonus as presented by the cases from the other studies.

Our patients developed a mild phenotype of CLN8 disease: as they presented mild epilepsy, cognitive decline, mild learning disability, attention-deficit/hyperactivity disorder (ADHD), they developed a markedly protracted course of motor decline. On the other hand, mild cerebellar atrophy was noted, no signs of visual failures, no loss of ambulation but, only one patient reaching ataxia at adolescence. This mild type of CLN8 disease may be attributed to the compound heterozygous variant of mutation carried by the patients (i.e., c.766C > G, p.(Gln256Glu) and c.473A > G, p.(Tyr158Cys)). This is in comparison to the homozygous mutation (c.766C > G, p.(Gln256Glu)) found in the three previous patients who presented a moderate to severe form of the disease [16].

By using a model structure of the CLN8 protein and various bioinformatic tools, we analyzed the *CLN8* gene variants presented in Table 1. We focused on the point variants, whose effect on the protein is not obvious, compared to the effects of large deletions. The analysis suggested several mechanisms through which the point variants could diminish the biological activity of the protein. Most of the variants seem likely to act by compromising the structural integrity of regions within the protein. This in turn is expected to reduce the overall stability of the protein and render the protein less active to various degrees. More specifically, the above effect of the variants may occur either through the loss of favorable non-covalent interactions that hold together bundled helices or through the induction of atomic clashes and desolvation effects.

The Q256E variant, which was found in the current study, is particularly interesting. While the variant may act as the other variants, i.e., by inducing conformational changes in the protein that destabilize it, it may also act in a more direct manner. The mutated position faces a cavity within the protein that, based on structural similarity to other proteins, may function as a binding pocket for a negatively charged ligand or substrate, most likely a lipid molecule. In such a case, the replacement of the electrically neutral side chain of Q256 with the negatively charged side chain of glutamate is expected to significantly diminish the protein’s ability to bind the ligand/substrate, with obvious functional consequences.

As the patients in our previous studies [10,16] and the current patients are distantly related, additionally, as our society (the Arab society in Israel) is also characterized by a high rate of consanguineous marriages [29,30], the occurrence of CLN8 disease could be detected in more future generations. Therefore, we recommend performing carrier screening of the *CLN8* gene variant, as well as the implementation of genetic counseling as a preventive measure, particularly in this village.

The current study could have some limitations. Test results are interpreted in the context of clinical findings, family history, and other laboratory data. Only variations in genes potentially related to the proband’s medical condition are reported. Undoubtedly, WES results increase accessibility to clinical genetic testing; however, they have limitations. Therefore, we recommend undertaking Whole Genome Sequencing (WGS) testing for the CLN-affected patients in the future. The bioinformatic analysis is based on a model structure of CLN8. The model was predicted using AlphaFold, a tool known to have near-experimental accuracy. However, it is possible that some of the residue–residue interactions may have been captured inaccurately, and since the analyses we performed emphasize such interactions, this might lead to misleading interpretation of the variants’ effects.

## Figures and Tables

**Figure 1 genes-13-01393-f001:**
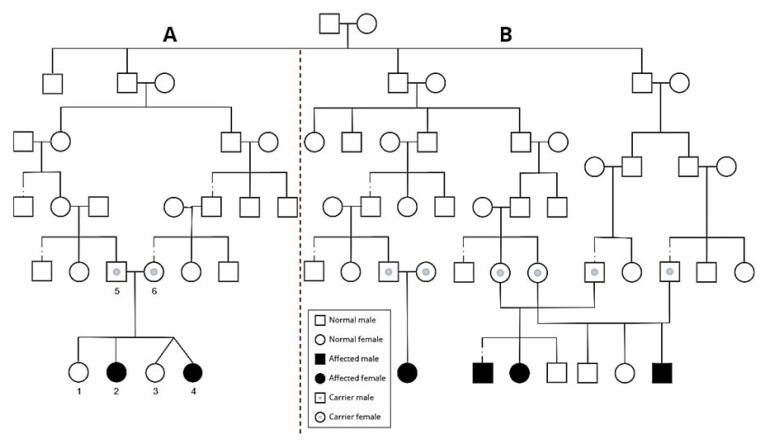
**Pedigree analysis of the family with *CLN8* variants.** Black circles (female) and squares (male) indicate affected family members, while open circles or squares indicate unaffected members. An open circle or square with a grey dot inside means the individual is a carrier. (**A**) Family pedigree of the current cases, including the two affected daughters (**2** and **4**), the two normal daughters (**1** and **3**), and their parents (**5** and **6**). Numbers 2 and 4 are compound heterozygous for the two variants; Tyr158Cys and Gln256Glu. The father (**5**) is heterozygous for Tyr158Cys, whereas the mother (**6**) is heterozygous for Gln256Glu. (**B**) The right side of the chart presents the previous cases. Affected individuals are homozygous for the variant Gln256Glu.

**Figure 2 genes-13-01393-f002:**
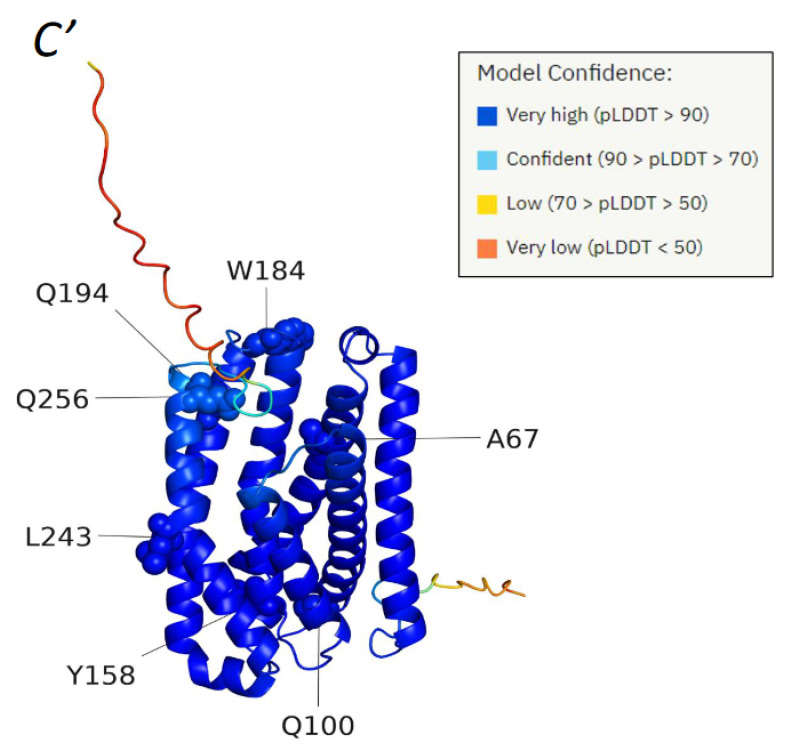
**Three-dimensional structure of CLN8′s protein product (NP_061764.2).** The predicted structure is shown as a ribbon and colored by AlphaFold’s confidence metric, pLDDT (see color code on the right). The residues targeted by the analyzed point mutations are shown as spheres and notated. The N and C termini are also notated.

**Figure 3 genes-13-01393-f003:**
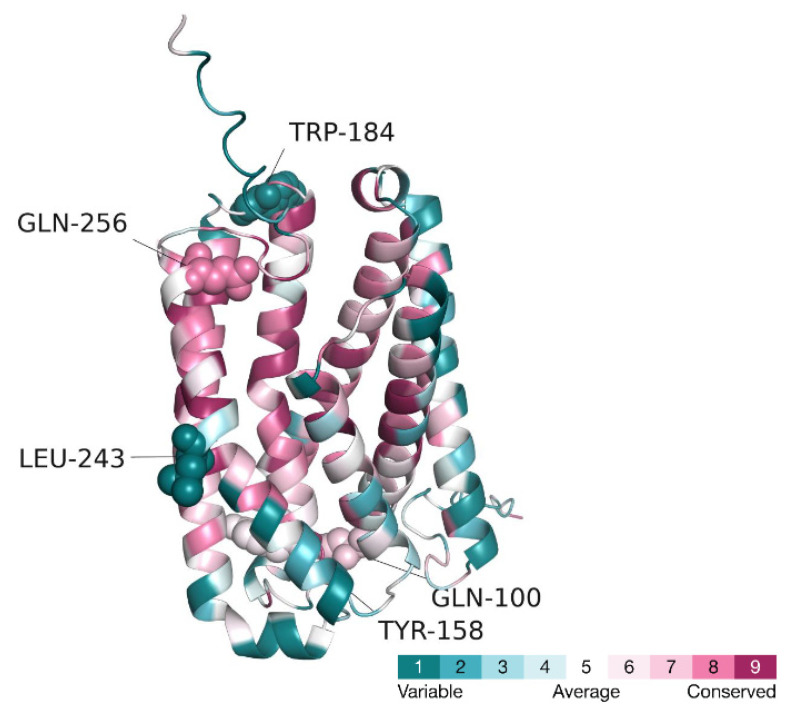
**Evolutionary conservation of the CLN8 protein.** The protein is presented as in Figure 1, except that it is colored by evolutionary conservation (see color code on the right). The conservation was calculated by the ConSurf server.

**Figure 4 genes-13-01393-f004:**
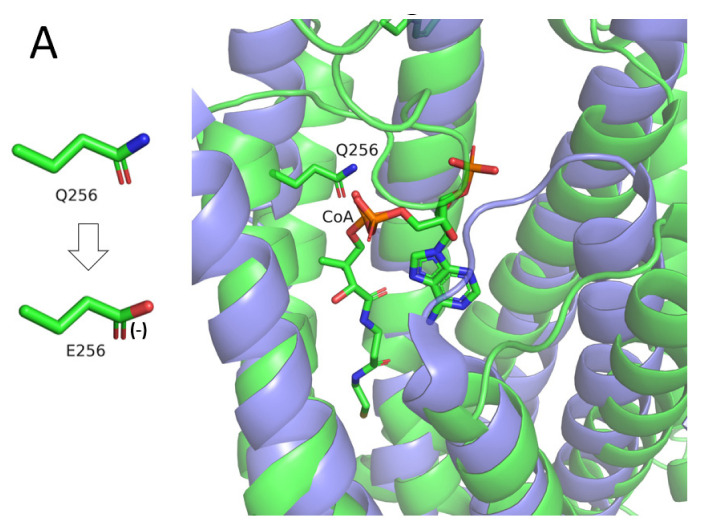
Superposition of (**A**) the mechanosensitive TACAN channel (blue) and (**B**) ELOVL fatty acid elongase 7 (cyan) onto CLN8 (green). The coenzyme A component of the ligand and Q256 are both shown as sticks. The change in Q256′s side chain following the variant is shown on the left.

**Figure 5 genes-13-01393-f005:**
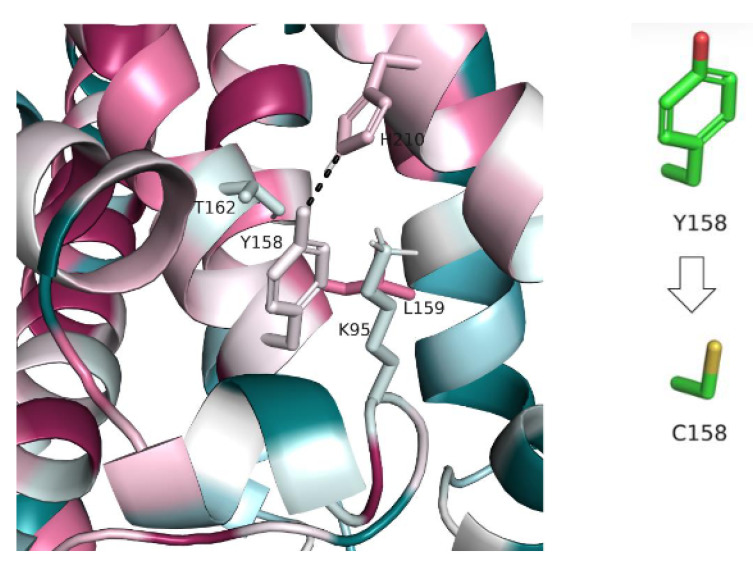
**Non-covalent interactions between Y158 and surrounding residues.** The protein is colored by conservation (see Figure 2). The interacting residues are shown as sticks and the hydrogen bond between Y158 and H210 is shown as a dashed line. The change in Y158′s side chain following the variant is shown on the left.

**Figure 6 genes-13-01393-f006:**
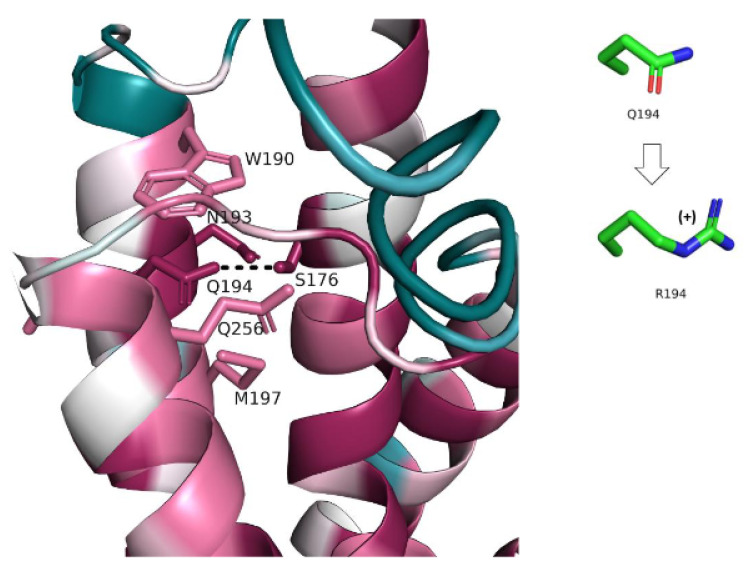
**Non-covalent interactions network involving Q194 and other evolutionarily conserved residues.** The protein is colored by conservation (see Figure 3). The interacting residues are shown as sticks and the hydrogen bond between Q194 and S176 is shown as a dashed line. The change in Q194′s side chain following the variant is shown on the left.

**Figure 7 genes-13-01393-f007:**
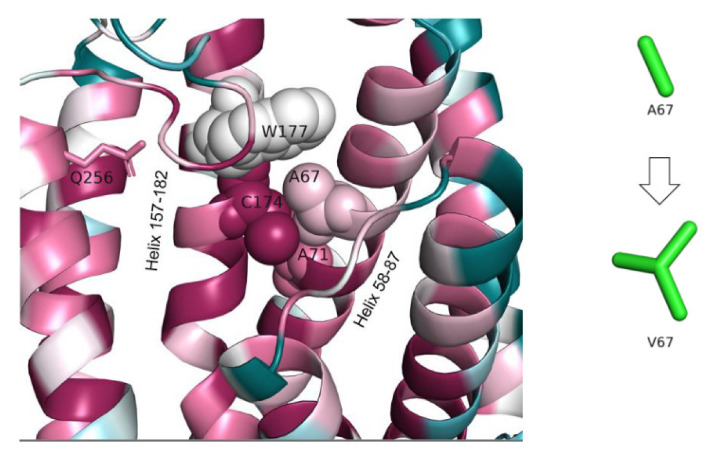
**Tight packing and non-covalent interactions between A67 on helix 58–87 and surrounding residues on helix 157–182.** The protein is colored by conservation (see Figure 3). The interacting residues are shown as spheres and the two helices are noted. The putative ligand/substrate-binding pocket is located left to helix 157–182. Q256, which might interact with the ligand/substrate from the opposite side of the pocket, is shown as well. The change in A67′s side chain following the variant is shown on the left.

**Figure 8 genes-13-01393-f008:**
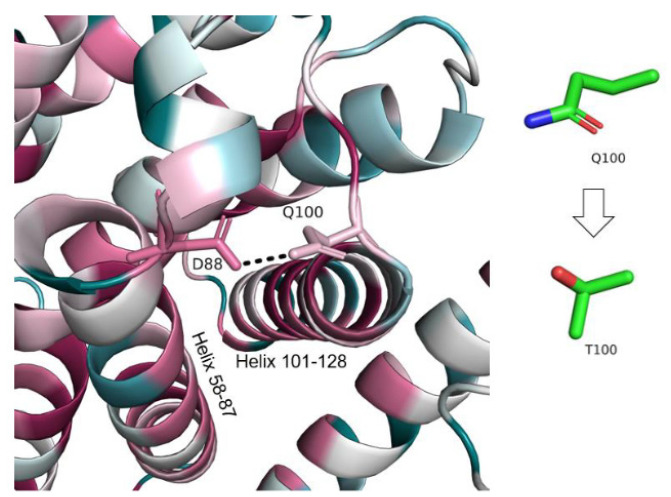
**Hydrogen bonding between Q100 and****D88.** The protein is colored by conservation (see Figure 3). The interacting residues are shown as sticks and the hydrogen bond is shown as a dashed line. The change in Q100′s side chain following the variant is shown on the left.

**Table 1 genes-13-01393-t001:** Summary of clinical phenotypes of our patients in comparison with cases from different studies having compound heterozygous variants in the *CLN8* gene.

The Study	Cannelli et al.’s Study in 2006	Allen et al.’s Study in 2012	Beesley et al.’s Study in 2016	Sanchez et al.’s Study in 2016	Gao et al.’s Study in2018	Our Study
*CLN8* Variant						
Allele-1	1-c.66delG, p.(Gly22Serfs*5) 2-c.66delG, p.(Gly22Serfs*5)	c.562_563delCT, p.(Leu188Valfs*58)	1–8p23.3 deletion, 54 Kb 2-c.763C > T, p.(Gln255*)	1-c.200C > T (p.A67V) 2-c.200C > T (p.A67V)	1-c.298C > T, p.(Gln100Ter)	1-c.473A > G, p.(Tyr158Cys) 2-c.473A > G, p.(Tyr158Cys)
Allele-2	1-c.473A > G, p.(Tyr158Cys) 2-c.581A > G, p.(Gln194Arg)	8p23.3 terminal deletion, de novo	1-c.728T > C, p.(Leu243Pro) 2–8p23.3 deletion, 235 Kb	1–8p23.3 deletion 2–8p23.3 deletion	1-c.551G > A, p.(Trp184Ter)	1-c.766C > G, p.(Gln256Glu) 2-c.766C > G, p.(Gln256Glu)
Clinical Features						
Number of patients	2	1	2	2	1	2
Consanguinity	No	No	No	NM	No	No
Country of origin	Italy	Ireland	United Kingdom	USA	China	Israel
Gender	M/ M	M	F/M	F/M	M	F/F
Disease onset: (y) (A) Onset abnormal gait (y)	3.5/4 3.5/4	4 4	4/ 3 (and 8m) 4 years 1 month/5 years and 2 months	7/5 NM	4 4	7.5/8 years 5 months No/12 years
(B) Onset visual impairment (y)	3.5/NM	Present at 5 years, no record of when the onset was	NM/at onset leading to failure and retinal dystrophy	14/5	7	No/at 11 years limitation of smooth pursuit (oculomotor apraxia). No visual impairment
(C) Onset myoclonus (y)	3.5/4 myoclonic jerks following febrile illness	Jerky ocular pursuit at 5 years no record of when the onset was	4 years 4 months (myoclonic seizure)/at age 7 years had hyperkinetic limb movements	NM	NM	No/no
(D) Onset epilepsy (y)	3/during 7 year follow up (myoclonic seizures)	4.5	4 years 4 months / 4 years 9 months	7/16	4	7.5/8 (and 5 m)
EEG	NM	Slow background, complex partial seizures	NM/abnormally slow background activity and runs of epileptiform discharges	NM	Irregular and slow background activity and high incidence of generalized sequences of atypical spike-wave discharges	Frontal sharp waves/normal
Loss of ambulation (y)	NM/during 7 year follow up	5.5	By 5 years 9 months/by 7 years	No/no	8 years six months	No/no
Visual failure (y)	3.5/during 7 year follow up	5.5	By 4 years 8 months/ 3 years 8 months	No/no	8 years six months	No/no
Previous developmental delay	Yes (motor and speech)/yes (motor)	Yes Global (motor and speech)	Mild speech delay/ no	No/no	No	Mild learning disability ADHD/ADHD
Head circumference (centile)	NM	50 cm (9th centile)	NM	NM	NM	50th percentile/ 40th percentile
Psychomotor regression	3.5/during 7 year follow up	Yes, at age 4 years	Yes, at onset/ yes, at 5 years and 2 months	No/no	Yes at 7 years old	Behavioral difficulties at age 11, speech disturbances at age 11 and 6 months/cognitive decline at 8 years 10 months
Brain imaging (MRI)	Cerebral and cerebellar atrophy/atrophied cerebellar vermis and cerebral cortex	Hyperintensity of PLIC, posterior dwm and centrum semi-oval (axial T2 FLAIR) plus cerebellar atrophy	Cerebellar atrophy, low signal change abnormality in thalami bilaterally/white matter abnormalities	NM	Diffuse cerebral and cerebellar atrophy	At age 8—mild cerebellar atrophy, MRI at age 12 years—worsening cerebellar atrophy/mild cerebellar atrophy, thalamic lesion

Abbreviations: F: female; M: male; NM: not mentioned; EEG: electroencephalography; ADHD: attention-deficit hyperactivity disorder; MRI: magnetic resonance imaging; Dwm: deep white matter; PLIC: posterior limb of the internal capsule.

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
