# Peer review of "CLN8 Gene Compound Heterozygous Variants: A New Case and Protein Bioinformatics Analyses"

_genes, 2022, doi:10.3390/genes13081393_

Round 1
Reviewer 1 Report
The authors describe two patients with atypical and protracted CLN8 disease carrying novel compound heterozygous mutation at the CLN8 gene has been identified. The authors explain the methodology and study design very well. In addition they performed bioinformatic analysis on the protein structure.
Author Response
Thank you for your kind response.
Reviewer 2 Report
Sharkia and colleagues describe two patients with compound heterozygous CLN8 variants who exhibit a milder NCL phenotype, as well as some analysis of how variants in the gene may affect protein structure and function. While this is valuable information, there are serious issues that must be addressed.
My biggest concern is with the variants themselves. The accession number provided (NM_001042432.1) is for CLN3, not CLN8. Additionally, I think that the variants are reported incorrectly and should be p.Q255E and p.Q256E. The variants being reported incorrectly makes me hesitant to analyze the rest of the data. Additionally, I don't think the authors make it clear that the p.Q256E variant has been published previously, and the ClinVar Variation ID # should be included.
I think addition of a pedigree would be very useful, especially if the authors can put the additional family members with the variant in. This would make it more clear where the variants come from, since they are so similar it gets confusing.
The authors state "bioinformatic analyses" in the title, but I would like to see more than the AlphaFold analysis (maybe add clustering analysis, MetaDome, REVEL, etc). Additionally, with the AlphaFold analysis, I'd like to know why they chose the variants they did. In HGMD there appear to be quite a few variants. Also, the p.W184Ter variant is analyzed as p.W184T, which isn't relevant. The variants should also be grouped by patient. Finally, the deletions are described as if they result in a truncated protein. Is there any evidence of nonsense mediated decay being escaped?
A figure showing all the reported variants would be nice.
A few smaller things: both CLN8 disease and NCL8 are used, and it isn't explicitly stated that variants in CLN8 result in NCL8. Just use one and be consistent.
Line 90 says that the patient history was remarkable when it should be "unremarkable."
Line 110/111: this should read: “Later, her first seizure…”
Line 116: I don’t think tandem needs to be capitalized.
Figure 1: this should read “three-dimensional structure of CLN8” or “…of CLN8’s…” Also the protein accession number (NP…) should be included.
Also, I think the figures could be consolidated. I was unable to look at the supplementary data due to file type, but some of the other variants AlphaFold information could be there.
Overall, this paper has value, but in its current form is not correct or clear enough to be useful. While these are small changes, I'm hesitant to recommend acceptance of the manuscript due to the type of errors I found.
Author Response
We thank you for your constructive criticism of our research work. This is clearly illustrated from your valuable comments, suggestions and recommendations which will definitely improve upon and contribute to the quality of our article.
Our responses to your valued points are as follows:
Reviewer 2:
My biggest concern is with the variants themselves. The accession number provided (NM_001042432.1) is for CLN3, not CLN8. Additionally, I think that the variants are reported incorrectly and should be p.Q255E and p.Q256E. The variants being reported incorrectly makes me hesitant to analyze the rest of the data. Additionally, I don't think the authors make it clear that the p.Q256E variant has been published previously, and the ClinVar Variation ID # should be included.
This point has caused a deep concern in us, which demanded a thorough investigation of all the available and relevant source data and files especially from the genetic point of view. Surprisingly, we found a contradiction between the exact gene mutation (in the genetic laboratory result) and the one reported to us. The same, has been corrected and the manuscript has been updated accordingly, therefore, we are grateful to you for your important and highly relevant note.
- The (NM_001042432.1) has been corrected to (3),
- The mutation c.763C>G, p.(Gln256Glu) has been corrected and changed to 473A>G, p.(Tyr158Cys) throughout the manuscript.
I think addition of a pedigree would be very useful, especially if the authors can put the additional family members with the variant in. This would make it more clear where the variants come from, since they are so similar it gets confusing.
Response: We appreciate your suggestion of this point which indeed improved the presentation of the relationship between the previous and current cases. Thus, a pedigree analysis was made and presented as Figure 1. It definitely added to the quality of the manuscript.
The authors state "bioinformatic analyses" in the title, but I would like to see more than the AlphaFold analysis (maybe add clustering analysis, MetaDome, REVEL, etc). Additionally, with the AlphaFold analysis, I'd like to know why they chose the variants they did. In HGMD there appear to be quite a few variants. Also, the p.W184Ter variant is analyzed as p.W184T, which isn't relevant. The variants should also be grouped by patient. Finally, the deletions are described as if they result in a truncated protein. Is there any evidence of nonsense mediated decay being escaped?
Response:
- We added a more detailed description of the protein structural bioinformatic methods to the Methods section (subsection 2.3). These include the structure prediction algorithm AlphaFold, the evolutionary conservation method ConSurf, the stability prediction method DUET, and the structural analogues scanning method Dali.
- We did not use methods like MetaDome, REVEL, MutPred, and PolyPhen, which predict which mutations in a gene are pathogenic, because we focused on mutations in CLN8 that our study and the other studies mentioned in Table 1 have already shown to be pathogenic. Our goal was to find the most probable molecular basis for the mutations’ pathogenicity, at the protein level. To this end, we used the protein structural bioinformatics methods mentioned above.
- We removed the analysis of the W184T mutation and added an analysis of W184Ter.
- We grouped the mutations according to the studies reporting them.
A figure showing all the reported variants would be nice.
Response: based on our bioinformatic methods, the supplied figures are obtained with the best possible clarity.
A few smaller things: both CLN8 disease and NCL8 are used, and it isn't explicitly stated that variants in CLN8 result in NCL8. Just use one and be consistent.
Response: the abbreviation NCL8 has been changed throughout the manuscript to CLN8 to remain consistent, as you suggested.
Line 90 says that the patient history was remarkable when it should be "unremarkable."
Response: the term "remarkable" has been changed to "unremarkable", as you demanded.
Line 110/111: this should read: “Later, her first seizure…”
Response: the word "her" has been added, as you requested.
Line 116: I don’t think tandem needs to be capitalized.
Response: the word has been corrected, as you suggested.
Figure 1: this should read “three-dimensional structure of CLN8” or “…of CLN8’s…” Also the protein accession number (NP…) should be included.
Response: The title was corrected and the NP number was added, as you requested.
Also, I think the figures could be consolidated. I was unable to look at the supplementary data due to file type, but some of the other variants AlphaFold information could be there.
Response: The AlphaFold data information shown in Figure 2 applies to all other figures as well. We did not model the variants as separate structures.
Overall, this paper has value, but in its current form is not correct or clear enough to be useful. While these are small changes, I'm hesitant to recommend acceptance of the manuscript due to the type of errors I found.

Reviewer 3 Report
In this study Sharkia R. et al. report the case of two female sibilings presented with atypical phenotypic manifestation and protracted clinical course of NCL8 carrying a novel compound heterozygous mutation at the CLN8 gene. Moreover, bioinformatics analyses of the CLN8 protein structure are performed and putative mutations consequences are discussed.
Although I consider the purpose of the manuscript interesting and in line with the Special Issue, I find the article lacking key methodological details.
Below find my comments:
Minor:
- - In the title I suggest to use “protein bioinformatics analyses” instead of “protein’s bioinformatic analyses”.
- - In line 27 (Abstract), no real “simulations” (i.e. molecular docking or molecular dynamics simulations) were performed.
- - In line 35 (Introduction), you should use “lipofuscinosis” instead of “lipofuscinoses”.
- - In lines 54-56 (Introduction), please revise the structure of this sentence.
- - Table 1 does not have a caption.
- - In the paragraph 3.1 Mutational analysis of the CLN8 gene product (Results), both title and text end with “:”.
- - In lines 166 (Results), there should be a typo in “Q296”.
- - In lines 175-176 ( caption of Figure 3) , there is a duplicated sentence.
- - In lines 270 (Discussion), I suggest to use “Compared” instead of “Comparative”.
- - Please revise all “…‘s” in the whole text (e.g. lines 26, 74, 139, 175, 190, 210, 232, 248).
Major:
- - In Materials and Methods section, the “Genetic Analysis” paragraph lacks important details about how sequencing and mutations identification were performed. Additionally, in lines 120-121 (“DNA was extracted from peripheral blood lymphocytes of all family members using standard procedures”) any reference of the “standard procedures” is missing.
- - In Materials and Methods section, a paragraph regarding the bioinformatics analyses, relative tools and parameters used is completely absent.
- - Throughout the manuscript (e.g. Table 1, third row (CLN8 mutation Allele-1) last column (Our study), and in lines 125-126 “missense mutation (c.763C>G, p.(Gln256Glu))”), I found an unclear key aspect: c.763C>G and c.766C>G could not result in the same p.(Gln256Glu) substitution. Aminoacidic residue Gln256 is encoded by codon 766-768, so the mutation c.763C>G should correspond to protein change p.(Gln255Glu), that is not considered in the study and, in particular, in the bioinformatics analyses.
- - Finally, in the Discussion section I suggest to add a paragraph dedicated to the limitations of the study.
Author Response
We thank you for your constructive criticism of our research work. This is clearly illustrated from your valuable comments, suggestions and recommendations which will definitely improve upon and contribute to the quality of our article.
Our responses to your valued points are as follows:
Reviewer 3:
In this study Sharkia R. et al. report the case of two female sibilings presented with atypical phenotypic manifestation and protracted clinical course of NCL8 carrying a novel compound heterozygous mutation at the CLN8 gene. Moreover, bioinformatics analyses of the CLN8 protein structure are performed and putative mutations consequences are discussed.
Although I consider the purpose of the manuscript interesting and in line with the Special Issue, I find the article lacking key methodological details.
Below find my comments:
Minor:
- - In the title I suggest to use “protein bioinformatics analyses” instead of “protein’s bioinformatic analyses”.
Response: the terms "protein bioinformatics analyses" has been corrected, as you suggested.
- - In line 27 (Abstract), no real “simulations” (i.e. molecular docking or molecular dynamics simulations) were performed.
Response: The term was corrected in the Abstract and in the Introduction.
- - In line 35 (Introduction), you should use “lipofuscinosis” instead of “lipofuscinoses”.
Response: the term has been corrected as suggested.
- - In lines 54-56 (Introduction), please revise the structure of this sentence.
Response: The sentence was restructured as requested.
- - Table 1 does not have a caption.
Response: a caption for the table has been added as required.
- - In the paragraph 3.1 Mutational analysis of the CLN8 gene product (Results), both title and text end with “:”.
Response: the punctuation marks ":" have been added after each sub-title of the "Results" section as suggested.
- - In lines 166 (Results), there should be a typo in “Q296”.
Response: the number was corrected.
- - In lines 175-176 ( caption of Figure 3) , there is a duplicated sentence.
Response: the repetition was removed.
- - In lines 270 (Discussion), I suggest to use “Compared” instead of “Comparative”.
Response: the word has been changed, as you suggested.
- - Please revise all “…‘s” in the whole text (e.g. lines 26, 74, 139, 175, 190, 210, 232, 248).
Response: the corrections had been implemented wherever applicable.
Major:
- - In Materials and Methods section, the “Genetic Analysis” paragraph lacks important details about how sequencing and mutations identification were performed. Additionally, in lines 120-121 (“DNA was extracted from peripheral blood lymphocytes of all family members using standard procedures”) any reference of the “standard procedures” is missing.
Response: the relevant details had been added as you suggested.
- - In Materials and Methods section, a paragraph regarding the bioinformatics analyses, relative tools and parameters used is completely absent.
Response: the relevant details had been added as you suggested.
- - Throughout the manuscript (e.g. Table 1, third row (CLN8 mutation Allele-1) last column (Our study), and in lines 125-126 “missense mutation (c.763C>G, p.(Gln256Glu))”), I found an unclear key aspect: c.763C>G and c.766C>G could not result in the same p.(Gln256Glu) substitution. Aminoacidic residue Gln256 is encoded by codon 766-768, so the mutation c.763C>G should correspond to protein change p.(Gln255Glu), that is not considered in the study and, in particular, in the bioinformatics analyses.
Response: This point has caused a deep concern in us, which demanded a thorough investigation of all the available and relevant source data and files especially from the genetic point of view. Surprisingly, we found a contradiction between the exact gene mutation (in the genetic laboratory result) and the one reported to us. The same, has been corrected and the manuscript has been updated accordingly, therefore, we are grateful to you for your important and highly relevant note.
- - Finally, in the Discussion section I suggest to add a paragraph dedicated to the limitations of the study.
Response: a paragraph concerning the limitations of the study had been added as you suggested.

Round 2
Reviewer 2 Report
The authors have updated the manuscript so it is much clearer! Thank you!
Just a few small things:
Abstract, line 23-24 should read "In the current study we describe two patients who presented with atypical phenotypic..."
Line 71 should say homozygous not homozygote. Also include the accession number for the variant (NM...). Line 72 should read "all homozygous for the same..."
Line 75 you should state what the novel variant is.
Line 87 replace "non-relative" with unrelated.
I think this is just a formatting issue, but the legend for Table 1 is just in the manuscript text. Also some of the numbers are smaller in the table.
CLN8 should be in italics when referencing the gene.
Line 262: the c.763C>T is a nonsense variant, not a deletion. Its likely due to nonsense mediated decay or nonstop RNA decay that its not expressed, not the loss of the ER retrieval signal.
Author Response
The authors have updated the manuscript, so it is much clearer! Thank you!
Again, we thank you for your great concern and the constructive criticism of our research work.
Just a few small things:
Abstract, line 23-24 should read "In the current study we describe two patients who presented with atypical phenotypic..."
Response: The word "who" has been added, as you requested.
Line 71 should say homozygous not homozygote. Also include the accession number for the variant (NM...). Line 72 should read "all homozygous for the same..."
Response: These suggested corrections have been implemented, as you requested.
Line 75 you should state what the novel variant is.
Response: The novel compound variant has been added, as you requested.
Line 87 replace "non-relative" with unrelated.
Response: The term "non-relative" has been replaced with "unrelated", as you demanded.
I think this is just a formatting issue, but the legend for Table 1 is just in the manuscript text. Also some of the numbers are smaller in the table.
Response: The format has been adjusted accordingly.
CLN8 should be in italics when referencing the gene.
Response: The abbreviation "CLN8" has been changed to italics wherever it indicates the gene.
Line 262: the c.763C>T is a nonsense variant, not a deletion. Its likely due to nonsense mediated decay or nonstop RNA decay that its not expressed, not the loss of the ER retrieval signal.
Response: We agree with you that the title is a bit misleading, thus we changed it to:
- Polypeptide chain termination after position 255 (c.763C>T).
We think that our explanation is more suitable and more clarification was added as follows: "the presence of nonsense mutation at position c.763C>T (codon 255) leads to premature chain termination. Accordingly, all of the following amino acids (codon 256-286) won’t be synthesized. Most importantly, the codons 283-286 (Lys-Lys-Arg-Pro; ER-retrieval signal) won’t be expressed. Therefore, the mutant won’t reach the ER."

Reviewer 3 Report
All points have been adequately addressed. Therefore, I have no further comments on current version of the manuscript. I would only suggest to remove all “:” at the end of paragraph titles.
Author Response
Reviewer 3:
All points have been adequately addressed. Therefore, I have no further comments on current version of the manuscript. I would only suggest to remove all “:” at the end of paragraph titles.
We thank you again for your great concern and the constructive criticism of our research work. All “:” at the end of paragraph titles were removed as you suggested.